# 2-Oxabicyclo[2.2.2]octane as a new bioisostere of the phenyl ring

Vadym V. Levterov[1], Yaroslav Panasiuk[1], Kateryna Sahun[1], Oleksandr Stashkevych[1], Valentyn Badlo[1], Oleh Shablykin[1,2], Iryna Sadkova[1], Lina Bortnichuk[1], Oleksii Klymenko-Ulianov[1], Yuliia Holota[1], Leonid Lachmann[3], Petro Borysko[1], Kateryna Horbatok[1], Iryna Bodenchuk[1], Yuliia Bas[4], Dmytro Dudenko[1] & Pavel K. Mykhailiuk [1]✉

The phenyl ring is a basic structural element in chemistry. Here, we show the design, synthesis, and validation of its new saturated bioisostere with improved physicochemical properties – 2-oxabicyclo[2.2.2]octane. The design of the structure is based on the analysis of the advantages and disadvantages of the previously used bioisosteres: bicyclo[1.1.1]pentane, bicyclo[2.2.2] octane, and cubane. The key synthesis step is the iodocyclization of cyclohexane-containing alkenyl alcohols with molecular iodine in acetonitrile. 2-Oxabicyclo[2.2.2]octane core is incorporated into the structure of Imatinib and Vorinostat (SAHA) drugs instead of the phenyl ring. In Imatinib, such replacement leads to improvement of physicochemical properties: increased water solubility, enhanced metabolic stability, and reduced lipophilicity. In Vorinostat, such replacement results in a new bioactive analog of the drug. This study enhances the repertoire of available saturated bioisosteres of (hetero)aromatic rings for the use in drug discovery projects.

The phenyl ring is a basic structural element in chemistry. It is one of the most common structural motifs in natural products[1] and bioactive compounds[2,3]. Moreover, more than five hundred drugs contain a fragment of *para*-substituted phenyl ring (Fig. 1a, b)[4], including the well-known to everyone Paracetamol. However, organic compounds with more than two phenyl rings often suffer from poor solubility[5–7].

In 2012, however, Stepan and colleagues showed that a replacement of the central phenyl ring in a γ-secretase inhibitor with the bicyclo[1.1.1]pentane improved its physicochemical properties and retained bioactivity[8–11]. Later, analogous replacements were undertaken with cubane[12–19], and bicyclo[2.2.2]octane (Fig. 1a, b)[20–22]. Therefore, during the past decade, these scaffolds proved to be useful in drug discovery, medicinal chemistry, and supramolecular chemistry[23–31]. Replacement of the *ortho*- and *meta*-substituted phenyl rings in bioactive compounds with saturated bioisosteres was also recently achieved[26–31]. Recent studies, however, showed that all three

bioisosteres had drawbacks. In bicyclo[1.1.1]pentane, the most popular among them today[32–40], the distance between two bridgehead carbon atoms (C-C) is 1.8 Å, which is ca. 35% shorter than that in the *para*-substituted phenyl ring (2.8 Å). Bicyclo[2.2.2]octane has a closer C-C distance (2.6 Å), but higher lipophilicity[41]. Cubane, in turn, was recently demonstrated to be unstable under contact with transition metals[42,43], under mechanochemical treatment or heating[44].

In this work, we have rationally designed, synthesized, and characterized the new bioisostere of the phenyl ring – 2-oxabicyclo[2.2.2] octane (Fig. 1c).

Interestingly, 2-oxabicyclo[2.2.2]octane core has been known in the literature, but not in the context of phenyl bioisostere. Chemists used it as a starting material in organic synthesis;[45,46] and in medicinal chemistry[47–49] as an analog of 4-aminopiperidine[50–53] or cyclohexane[54,55]. Also, 2-oxabicyclo[2.2.2]octane containing molecules exhibited a broad range of biological activities: estrogen receptor-beta

[1]Enamine Ltd., Winston Churchill street 78, 02094 Kyiv, Ukraine. [2]V. P. Kukhar IBOPC of the NASciences of Ukraine, Academician Kukhar Str. 1, 02094 Kyiv, Ukraine. [3]Bienta, Winston Churchill street 78, 02094 Kyiv, Ukraine. [4]Taras Shevchenko National University of Kyiv, Chemistry Department, Volodymyrska 64, 01601 Kyiv, Ukraine. ✉e-mail: Pavel.Mykhailiuk@gmail.com

**Fig. 1 | The *para*-substituted phenyl ring and its saturated bioisosteres. a** The *para*-substituted phenyl ring is a part of >500 drugs and agrochemicals. Bicyclo[1.1.1]pentanes, bicyclo[2.2.2]octanes, and cubane as saturated bioisosteres of the *para*-substituted phenyl ring. **b** Bioactive derivatives of bicyclo[1.1.1]pentanes, bicyclo[2.2.2]octanes, and cubane are described in >3000 patents. **c** Aim of this work: replacement of the *para*-substituted phenyl ring in bioactive compounds with 2-oxabicyclo[2.2.2]heptane. **d** Previous syntheses of 2-oxabicyclo[2.2.2]heptane by Singh, Fukuda (2014)[50] and Harrison (2019)[54].

agonists[47], myeloperoxidase inhibitors[48], antibacterial agents[49–53], DGAT1 Inhibitors[54], and RORγt agonists[55].

## Results

### Design

In the design of the improved phenyl bioisostere, we first needed to keep the advantages of the previously used cores: their conformational rigidity, metabolic stability, non-chirality, and collinearity of vectors ($\varphi = 180°$). At the same time, we needed to address their drawbacks: C-C distance, and lipophilicity. Considering the possible saturated structures (for the details of the design, please, see Supplementary Iinformation, page 5, Supplementary Fig. 1.), we decided to select the bicyclo[2.2.2]octane scaffold, because of its appropriate C-C distance, and decorate it with an oxygen atom. In particular, replacing one carbon atom with oxygen would give 2-oxabicyclo[2.2.2]octane with similar geometry and reduced lipophilicity (Fig. 1c). Also, this structure should be chemically stable as a simple derivative of tetrahydropyran.

### Optimization

Synthesis of the 2-oxabicyclo[2.2.2]octane core has been previously reported. In 2014, Singh and Fukuda obtained compound **1** from diethyl malonate (**2**) in 15 steps using alkylation as a key reaction (Fig. 1d)[50]. In 2019, Harrison synthesized compound **3** from ester **4** in six steps employing an intramolecular Michael addition (Fig. 1d)[54]. The latter approach was limited only to aromatic substituents. We, however, needed a general modular method that would give 2-oxabicyclo[2.2.2]octanes with one or two functional groups that could be subsequently modified to obtain a wide variety of derivatives - bioisosteres of the *mono-* and *para*-substituted phenyl rings.

Previously, we showed that smaller 2-oxabicyclo[2.1.1]hexane could be assembled via the iodocyclization reaction of the corresponding cyclobutane alkenyl alcohols[56]. The reaction proceeded with I₂/NaHCO₃ in the mixture of water and MeOtBu at room temperature. We hoped that similar cyclization would also take place with cyclohexane **5** (please, see its preparation below). However, under

analogous conditions the expected product **6** was not formed (Table 1, entry 1). We repeated the reaction several times varying the time and the temperature, however, with the same negative outcome (Table 1, entries 2-4). The addition of the iodine molecule to the double C=C bond did take place, but the cyclization failed to occur.

Subsequently, we realized that in contrast to the already conformationally preorganized small cyclobutane, the larger and more flexible cyclohexane ring should adopt the highly energetic boat

### Table 1 | Optimization of the synthesis of compound 6

| Entry | Conditions | Yield (%)[a] |
|---|---|---|
| 1 | I₂, NaHCO₃, MeOtBu, H₂O, rt, 12 h | n.d. |
| 2 | I₂, NaHCO₃, MeOtBu, H₂O, rt, 48 h | n.d. |
| 3 | I₂, NaHCO₃, MeOtBu, H₂O, rt, 1 h | n.d. |
| 4 | I₂, NaHCO₃, MeOtBu, H₂O, reflux, 12 h | n.d. |
| 5 | I₂, NaHCO₃, Et₂O, H₂O, rt, 12 h | n.d. |
| 6 | I₂, NaHCO₃, dioxane, H₂O, rt, 12 h | n.d. |
| 7 | I₂, NaHCO₃, dioxane, rt, 12 h | n.d. |
| 8 | I₂, NaHCO₃, MeOtBu, rt, 12 h | n.d. |
| 9 | I₂, NaHCO₃, DMF, rt, 12 h | <10 |
| 10 | I₂, NaHCO₃, DMSO, rt, 12 h | <10 |
| 11 | I₂, NaHCO₃, NMP, rt, 12 h | <10 |
| 12 | I₂, NaHCO₃, CH₃CN, rt, 12 h | 56 |
| 13 | I₂, NaHCO₃, CH₃CN, reflux, 12 h | 45 |
| 14 | Br₂, NaHCO₃, CH₃CN, rt, 12 h | 30 |

*N.d.* not determined.
[a]Isolated yield.

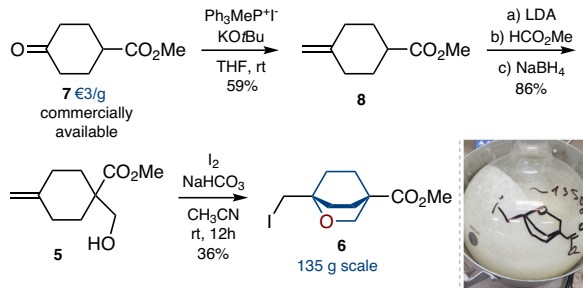

**Fig. 2 | Scalable synthesis of 2-oxabicyclo[2.2.2]octane 6.** The synthesis started from the commercially available ketone 7. Iodide 6 was obtained on a 135 g scale in one run.

conformation first (Table 1). The resulting entropic penalty seems to prevent the cyclization from occuring. We also tried other combinations of solvents with no success, however (Table 1, entries 5-8). Finally, we used solely dipolar aprotic solvents. Indeed, in dimethyl formamide, the formation of traces of the needed product was finally seen (Table 1, entry 9). A similar result was observed in dimethyl sulfoxide and N-methyl pyrrolidone (Table 1, entries 10, 11). In acetonitrile, the transformation proceeded cleaner, and iodide 6 was obtained in a 56% yield (Table 1, entry 12). Performing the reaction under heating (Table 1, entry 13) or employing bromine (Br₂; Table 1, entry 14) did not improve the yield.

## Scalable synthesis

Having a working procedure in hand, we studied its scalability. The whole synthesis scheme commenced from the commercially available ketoester 7 (ca. 3€/g, Fig. 2). Wittig reaction gave alkene 8 in 59% yield. Treatment of the latter with LDA/methyl formate followed by the reduction of the intermediate aldehyde with NaBH₄ gave alcohol 5 in 86% combined yield. Finally, the key iodocyclization was attempted on a multigram scale. Pure iodide 6 was obtained as a white crystalline solid after column chromatography with a 36% yield. Despite a rather moderate yield, this protocol allowed us to obtain 135 g of the product in a single run.

## Scope

Next, we studied the generality of the developed protocol. Treatment of alkene 8 with LDA/acetaldehyde gave the intermediate alcohol that was used in the subsequent iodocyclization under the developed conditions. The expected iodide 9 was isolated in 50% yield after column chromatography (Fig. 3a). Initially, we isolated the intermediate alcohol, but subsequently, we understood that performing the two-step procedure with a simple solvent swap ensured a better yield of the final product.

The reaction with aliphatic (10–12), aromatic (13–17), and heterocyclic (18–27) aldehydes gave the corresponding 2-oxabicyclo[2.2.2]octanes in good yields. Functional groups such as nitro, trifluoromethoxy, trifluoromethyl, nitrile, and halogen atoms tolerated the reaction conditions. The protocol was not without limitations, however. We could not obtain products 28, and 29 with thiazole and triazole heterocycles, due to the formation of complex mixtures (Fig. 3a). Ketones could also be used as electrophiles instead of aldehydes. As a representative example, the reaction of alkene 8 with LDA/acetone followed by iodocyclization gave dimethyl-substituted product 30 in 81% yield. The structure of 30 was confirmed by X-ray crystallographic analysis (Fig. 3b, Supplementary Data 1). A reduction of 8 followed by iodocyclization gave iodide 31 in 58% yield. Interestingly, the cyclization was not efficient at room temperature, and required heating. Alkylation of 8 with MeI or BnOCH₂Cl followed by reduction and iodocyclization gave the disubstituted products 32, 33 in 59-64% yield (Fig. 3b).

Trisubstituted exocyclic alkenes also afforded the desired 2-oxabicyclo[2.2.2]octane skeleton. For example, the iodocyclization of alkene 34 under the standard conditions gave iodide 35 in 67% yield (Fig. 3c). Alkene 36 provided iodide 37 in 50% yield. Endocyclic alkene 38, however, gave the isomeric core - 6-oxabicyclo[3.2.1]octane 39[57].

We also tried to assemble a 2-azabicyclo[2.2.2]octane skeleton using the developed strategy. An attempted iodocyclization of alkene 40 did not lead to the formation of the cyclic iodide 41 neither at room temperature nor under heating (Fig. 3d). However, the analogous reaction of the bridgehead-substituted alkene 42 at room temperature did give the needed product 43 in 31% yield. Under heating, the yield was improved to 41%.

## Modifications

Several representative modifications of the obtained iodides were undertaken to obtain various *mono-* and bifunctional 2-oxabicyclo[2.2.2]octanes ready for direct use in medicinal chemistry projects. Treatment of iodide 31 with potassium thioacetate followed by oxidation with NCS gave aliphatic sulfonyl chloride 44 in 85% yield. The reaction of 31 with potassium acetate and the subsequent alkali hydrolysis provided valuable alcohol 45. Oxidation of the latter afforded carboxylic acid 46 in 89% yield (Fig. 4).

Hydrogenative reduction of the C-I bond in iodide 6 followed by saponification of the ester group gave methyl acid 47. The Curtius reaction of the latter resulted in amine 48. The reaction of iodide 6 with LiAlH₄ gave alcohol 49 in 90% yield. O-Mesylation and the subsequent reaction with LiBr provided bromide 50. Swern oxidation of alcohol 49 gave aldehyde 51 in 63% yield. Isomeric methyl-substituted 2-oxabicyclo[2.2.2]octanes were obtained from iodide 32. Its reaction with sodium azide followed by the reduction formed amine 52. The reaction of iodide 32 with potassium acetate and hydrolysis gave alcohol 53 - isomer of alcohol 49. Oxidation of 53 formed carboxylic acid 54 - isomer of acid 47. Sulfonyl chloride 55 was also obtained from iodide 32 via a two-step procedure (Fig. 4).

From iodide 6 we also synthesized various bifunctional linkers for incorporation into bioactive compounds instead of the *para*-substituted phenyl ring. Saponification of ester 6 provided carboxylic acid 56 in 90% yield. The subsequent Curtius reaction afforded N-Boc iodide 57 in 87% yield. The structure of 57 was confirmed by X-ray crystallographic analysis (Supplementary Data 2). The reaction of the latter with potassium acetate, followed by ester hydrolysis (via 58) and N-Boc acidic deprotection gave amino alcohol 59. Oxidation of the alcohol group in 58 gave N-Boc protected amino acid 60 – a saturated analog of the *para*-aminobenzoic acid. The reaction of iodide 57 with NaN₃ (via azide 61) followed by reduction of the azide group formed diamine 62. The reaction of iodide 6 with NaN₃ (via azide 63), the subsequent reduction (via 64), N-Boc protection, and saponification gave another N-Boc protected amino acid 65. The Curtius reaction of the latter provided N-Boc diamine 66 – isomer of diamine 62. The reaction of iodide 6 with sodium azide followed by extensive reduction of the intermediate azide with LiAlH₄ gave amino alcohol 67. The structure of 67 was confirmed by X-ray crystallographic analysis (Supplementary Data 3). The reaction of iodide 6 with potassium acetate (via 68) followed by saponification of the ester group and oxidation gave linker 69. Its structure was also confirmed by X-ray crystallographic analysis (Supplementary Data 4). Worth noting that all the above-described syntheses depicted in Fig. 4 were realized on a multigram scale.

Alkylation of 4-bromothiophenol with iodide 32 followed by oxidation of the intermediate sulfide gave sulfone 70 in 54% yield over two steps (Fig. 4). Sulfonamide 71 was obtained in 44% yield from amine 48. Cu-catalyzed click reaction between azide 63 and 3-ethynylquinoline smoothly provided triazole 72. Condensation of acid 73 with N-hydroxyphthalimide (NHPI) in the presence of N,N'-diisopropylcarbodiimide (DIC) gave the activated ester 74 (Fig. 4). Ni-mediated

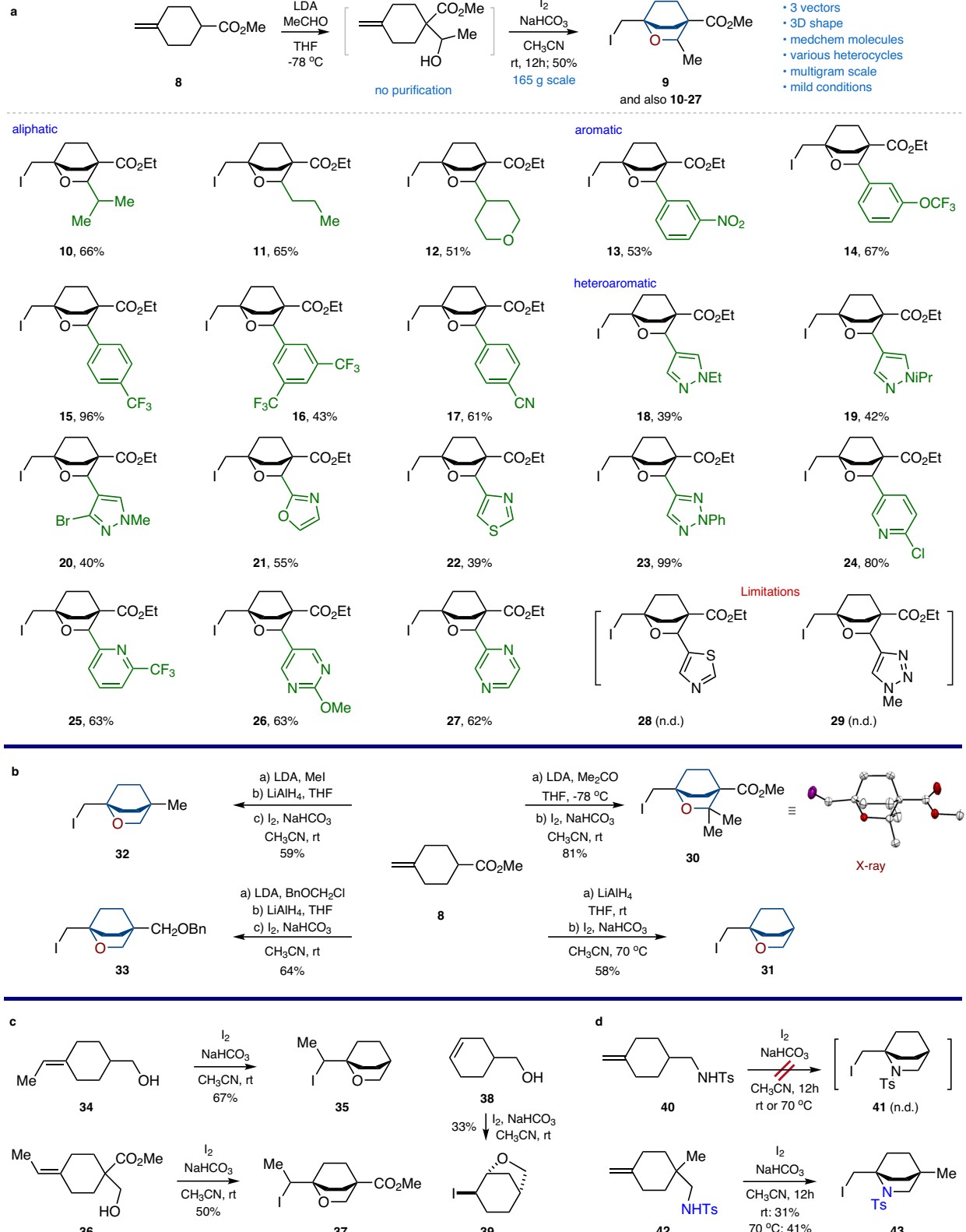

**Fig. 3 | Synthesis of 2-oxabicyclo[2.2.2]octanes and 2-azabicyclo[2.2.2]octanes.**
**a** Synthesis of 2-oxabicyclo[2.2.2]octanes with three exit vectors (for products
**10–29**, ethyl ester analog of alkene **8** was used). **b** Synthesis of 2-oxabicyclo[2.2.2]
octanes with one and two exit vectors. X-ray crystal structure of compound **30b**
(carbon – white, oxygen – red, iodine - violet). Hydrogen atoms are omitted for
clarity. **c** Iodocyclization of alkenes **34, 36,** and **38**. **d**, Synthesis of 2-azabicy-
clo[2.2.2]octane **43**.

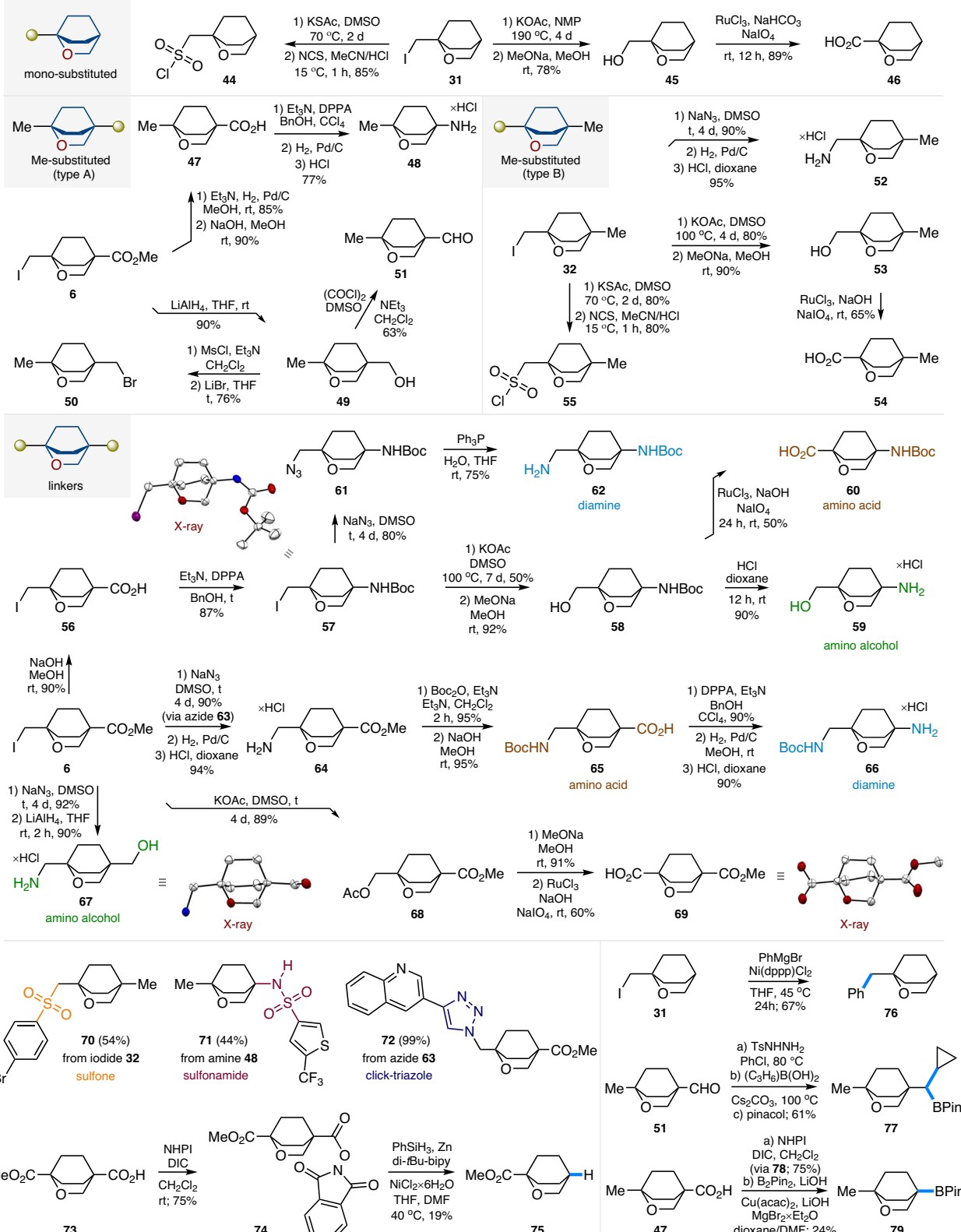

**Fig. 4 | Synthesis of functionalized 2-oxabicyclo[2.2.2]octanes for medicinal chemistry.** X-ray crystal structure of compounds **57**, **67**, and **69** (carbon – white, oxygen – red, nitrogen – blue, iodine - violet). Hydrogen and chlorine atoms are omitted for clarity.

Barton decarboxylation[58] of the latter with PhSiH₃ was performed next to provide ester **75**.

Ni-Mediated C-C cross-coupling of iodide **31** with PhMgBr gave 2-oxabicyclo[2.2.2]octane **76** in 67% yield. The reaction of aldehyde **51** with p-toluenesulfonyl hydrazide gave the

intermediate hydrazone that upon treatment with the cyclopropylboronic acid and pinacol provided organoboron derivative **77**[59]. Condensation of acid **47** with *N*-hydroxyphthalimide gave the activated ester **78**. Its structure was confirmed by X-ray crystallographic analysis (Supplementary Data 5). Cu-Catalyzed

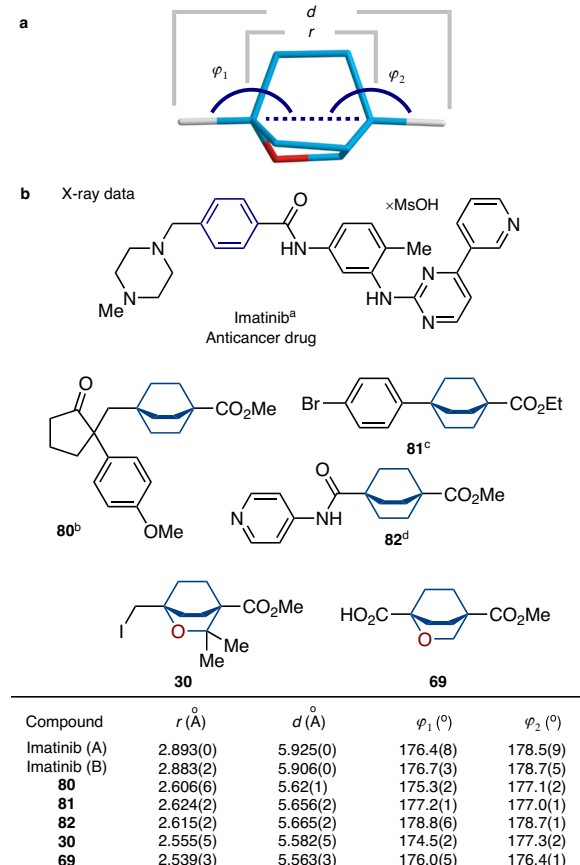

**Fig. 5 | Crystallographic analysis of 2-oxabicyclo[2.2.2]octanes. a** Definition of distances $r$, $d$ and angles $\varphi_1$, $\varphi_2$ (2-oxabicyclo[2.2.2]octane core is shown as example). **b** Geometric parameters $r$, $d$ and $\varphi_1$, $\varphi_2$ for *para*-substituted phenyl ring (Imatinib drug), its known saturated bioisosteres **80–82** and the new saturated bioisosteres **30, 69**. aData is taken from ref. 64. bData is taken from ref. 61. cData is taken from ref. 62. dData is taken from ref. 63. Two individual molecules of Imatinib (A and B) are present in the crystal lattice.

| Compound | $r$ (Å) | $d$ (Å) | $\varphi_1$ (°) | $\varphi_2$ (°) |
|---|---|---|---|---|
| Imatinib (A) | 2.893(0) | 5.925(0) | 176.4(8) | 178.5(9) |
| Imatinib (B) | 2.883(2) | 5.906(0) | 176.7(3) | 178.7(5) |
| **80** | 2.606(6) | 5.62(1) | 175.3(2) | 177.1(2) |
| **81** | 2.624(2) | 5.656(2) | 177.2(1) | 177.0(1) |
| **82** | 2.615(2) | 5.665(2) | 178.8(6) | 178.7(1) |
| **30** | 2.555(5) | 5.582(5) | 174.5(2) | 177.3(2) |
| **69** | 2.539(3) | 5.563(3) | 176.0(5) | 176.4(1) |

**Fig. 6 | Experimental pKa values of carboxylic acids 47, 54, 83, and 84.** Data is obtained by the titration method. *Para*-methyl benzoic acid (**83**) is used as a reference.

| | Carboxylic acid | pK$_a$ (exp.) |
|---|---|---|
| **83** | Me—⬡—CO$_2$H | 4.5 ± 0.1 |
| **84** | Me—⬡—CO$_2$H | 5.6 ± 0.1 |
| **47** | Me—⬡—CO$_2$H | 4.4 ± 0.1 |
| **54** | Me—⬡—CO$_2$H | 4.1 ± 0.1 |

decarboxylative borylation[60] of ester **78** gave organoboron derivative **79** (Fig. 4).

## Chemical stability

We also examined the thermal and chemical stability of the synthesized 2-oxabicyclo[2.2.2]octanes. As representative examples, we selected three molecules: isomeric acids **47, 54**, and amine **52**. All 2-

oxabicyclo[2.2.2]octanes were crystalline solids that were stable in air. We stored them in stock at room temperature in closed vials and observed no changes according to [1]H NMR after one year. Also, the compounds remained stable even under heating at 100 °C for five minutes. Treatment of the selected 2-oxabicyclo[2.2.2]octanes with aq. 1 M HCl, or aq. 1 M NaOH at room temperature for 1 h resulted in no decomposition either.

## Crystallographic analysis

Next, we compared the geometric properties of 2-oxabicyclo[2.2.2] octanes with those of the *para*-substituted phenyl ring, and the previously used bioisosteres - bicyclo[2.2.2]octanes. For this purpose, we measured two C-C distances $r$ and $d$ to see the overall similarity of cores; and two angles $\varphi_1$ and $\varphi_2$ to estimate the collinearity of exit vectors (Fig. 5a).

We calculated the values of $r$, $d$, $\varphi_1$, and $\varphi_2$ of 2-oxabicyclo[2.2.2] octanes from the X-ray data of compounds **30, 69**. The related parameters for bicyclo[2.2.2]octanes **80**[61], **81**[62], and **82**[63] were calculated from their X-ray data published in the literature (Fig. 5b). The corresponding parameters for the *para*-substituted phenyl ring were calculated from the reported crystal structure of the anticancer drug Imatinib[64]. Analysis of this data revealed that the geometric properties of 2-oxabicyclo[2.2.2]octanes were indeed very similar to those of the *para*-substituted phenyl ring. The distance r in 2-oxabicyclo[2.2.2] octanes was ca. 0.3 Å shorter than that in the *para*-phenyl ring: 2.54–2.56 Å vs 2.88–2.89 Å (*para*-phenyl). The distance d between substituents in 2-oxabicyclo[2.2.2]octanes was also ca. 0.3 Å shorter than that in the *para*-phenyl ring: 5.56–5.58 Å vs 5.90–5.93 Å (*para*-phenyl). The difference in collinearity of vectors was insignificant, as angles $\varphi_1$ and $\varphi_2$ were almost identical in both scaffolds: 176-177° vs 178-179° (*para*-phenyl). Interestingly, even in the *para*-substituted phenyl ring in Imatinib in the crystal phase, the observed angles $\varphi_1$ and $\varphi_2$ deviated from the ideal value of 180°: 176-179°. It must be noted, that all parameters, - $r$, $d$, $\varphi_1$ and $\varphi_2$, - were also almost identical in both bicyclo[2.2.2]octanes (**80–82**) and 2-oxabicyclo[2.2.2]octanes (**30, 69**) (Fig. 5b).

In short summary, the replacement of the methylene group for an oxygen atom in the bicyclo[2.2.2]octane core did not affect its three-dimensional geometry. Moreover, the formed 2-oxabicyclo[2.2.2] octane core resembled well the *para*-substituted phenyl ring, as the geometric parameters r, d, $\varphi_1$, and $\varphi_2$ remained very similar (please, see SI, page 277, Supplementary Fig. 8).

## The acidity of functional groups

We also studied the influence of the replacement of the methylene group for an oxygen atom in the bicyclo[2.2.2]octane skeleton on the electronic properties. Towards this goal, we measured experimental pK$_a$ values of isomeric 2-oxabicyclo[2.2.2]octane carboxylic acids **47** and **54**, bicyclo[2.2.2]octane carboxylic acid **84**, and *para*-methyl benzoic acid (**83**) as a reference (Fig. 6). Replacement of the methylene group in **84** for the oxygen atom at the distal γ-position notably increased its acidity from pK$_a$ = 5.6 to 4.4 (**47**). However, analogous replacement at the β-position increased the acidity even more to pK$_a$ = 4.1 (**54**).

Important to mention that the acidity of aromatic carboxylic acid **83** and 2-oxabicyclo[2.2.2]octane **47** were almost identical (Fig. 6). The replacement of the phenyl ring in acid **83** with the bicyclo[2.2.2]octane core reduced the acidity: pK$_a$ = 4.5 (**83**) vs 5.6 (**84**). However, incorporation of the β-oxygen atom into the latter ideally restored it: pK$_a$ = 4.4 (**47**). Because the acidity/basicity of functional groups is often responsible for the potency, selectivity, and toxicity of bioactive compounds[65], the fine-tuning of the pK$_a$ by replacing the phenyl ring with isomeric 2-oxabicyclo[2.2.2]octanes could become a preferred solution.

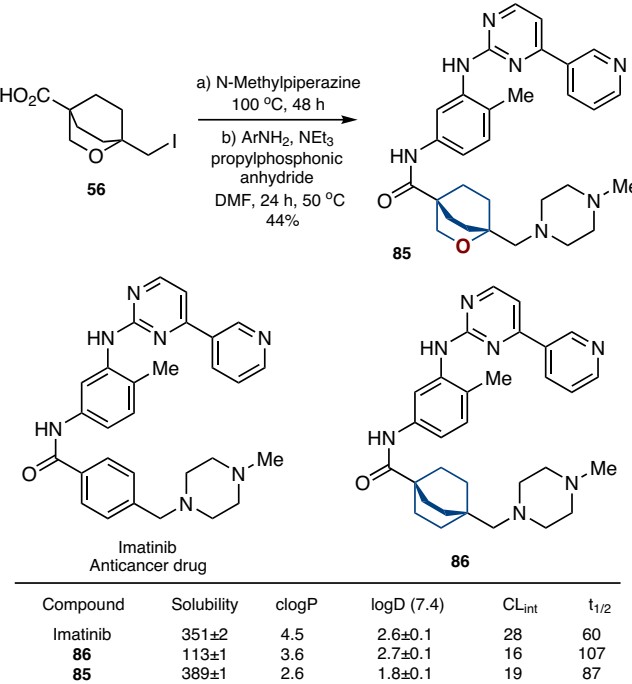

**Fig. 7 | Replacement of the *para*-phenyl ring with saturated bioisosteres in anticancer drug Imatinib.** Solubility: experimental kinetic solubility in phosphate-buffered saline, pH 7.4 (μM). clogP: calculated lipophilicity. logD (7.4): experimental distribution coefficient in *n*-octanol/phosphate-buffered saline, pH 7.4. Reliable logD measured were obtained within a range of 1.0–4.5. $CL_{int}$ clearance intrinsic: experimental metabolic stability in human liver microsomes (μl/min/mg). $t_{1/2}$ (min) experimental half-time of a metabolic decomposition.

| Compound | Solubility | clogP | logD (7.4) | $CL_{int}$ | $t_{1/2}$ |
|---|---|---|---|---|---|
| Imatinib | 351±2 | 4.5 | 2.6±0.1 | 28 | 60 |
| **86** | 113±1 | 3.6 | 2.7±0.1 | 16 | 107 |
| **85** | 389±1 | 2.6 | 1.8±0.1 | 19 | 87 |

**Fig. 8 | Replacement of the phenyl ring with saturated bioisosteres in anticancer drug Vorinostat (SAHA). a** Synthesis of compound **88** – a saturated analog of Vorinostat. Reaction conditions: a) Cl(O)C(CH₂)₆CO₂Me, NEt₃, CH₂Cl₂, rt, 2 h. b) NaOH, MeOH, reflux, 30 min. c) NH₂OH·HCl, DMF, CDI, rt, 30 min. **b** Structure of Vorinostat (SAHA), and its saturated analog **89**.

## Incorporation into drugs

To demonstrate the practical utility of the 2-oxabicyclo[2.2.2]octane scaffold, we incorporated it into the structure of anticancer drugs Imatinib, and Vorinostat (SAHA) instead of the *para*- and *mono*-substituted phenyl rings, correspondingly (Figs. 7 and 8).

The reaction of iodide **56** with *N*-methyl piperazine, followed by acylation with the substituted aniline gave compound **85** – a saturated analog of Imatinib (Fig. 7). For comparison, we also synthesized compound **86** with the bicyclo[2.2.2]octane core (please, see SI, pages 52-54). The commercialized drug Imatinib is used in practice as a mesylate salt. However, to estimate the impact of the replacement of the phenyl ring with bioisosteres on the physicochemical properties, we prepared and studied all three compounds, - **85**, **86**, Imatinib, - as free bases.

From amine **87**, in three steps we synthesized compound **88** - a saturated analog of Vorinostat (Fig. 8). For comparison, we also obtained analog **89** with the bicyclo[2.2.2]octane skeleton (please, see SI, pages 56, 57).

## Physicochemical properties

Replacement of the *para*-substituted phenyl ring in Imatinib by bicyclo[2.2.2]octane (**86**) decreased the water solubility by more than three times (Fig. 7). However, the incorporation of the 2-oxabicyclo[2.2.2] octane (**85**) in Imatinib increased the solubility close to the original values: 351 μM (Imatinib) vs 113 μM (**86**) vs 389 μM (**85**).

To study the replacement of the phenyl ring with saturated bioisosteres on lipophilicity, we used two characteristics: calculated (clogP)[66] and experimental (logD) lipophilicities. Incorporation of bicyclo[2.2.2]octane in sted of the phenyl ring resulted in a decrease of clogP: 4.5 (Imatinib) vs 3.6 (**86**). The incorporation of 2-oxabicyclo[2.2.2]octane led to an even further decrease of clogP: 2.6 (**85**). A somewhat similar trend was observed with the experimental lipophilicity, logD. While the incorporation of the bicyclo[2.2.2]octane core into Imatinib did not significantly affect it; incorporation of the 2-oxabicyclo[2.2.2]octane core reduced it by ca. one unit, logD: 2.6 (Imatinib) vs 2.7 (**86**) vs 1.8 (**85**).

The effect of saturated bioisosteres on metabolic stability was studied next. The incorporation of both bicyclo[2.2.2]octane (**86**) and 2-oxabicyclo[2.2.2]octane (**85**) into Imatinib, increased the metabolic stability in human liver microsomes: $CL_{int}$ (mg/(min•μL)) = 28 (Imatinib) vs 16 (**86**) vs 19 (**85**) (Fig. 7). Moreover, incorporation of the 2-oxabicyclo[2.2.2]octane core (**85**) into Imatinib increased the life half time by almost 50%: $t_{1/2}$ (min) = 60 (Imatinib) vs 87 (**85**).

In summary, the replacement of the *para*-substituted phenyl ring in Imatinib with common bicyclo[2.2.2]octane core (**86**) led to an undesired three-times decrease in water solubility. At the same time, analogous replacement with 2-oxabicyclo[2.2.2]octane (**85**) resulted in an improvement of all measured physicochemical parameters: increased solubility, enhanced metabolic stability, and reduced lipophilicity.

## Biological activity

Finally, to answer a key question, - whether the 2-oxabicyclo[2.2.2] octane core could indeed mimic the phenyl ring in bioactive compounds, we measured the biological activity of Imatinib versus its analogs **85**, **86**; and Vorinostat versus its analogs **88**, **89**.

We studied the inhibitory effect of Imatinib, Staurosporine, and compounds **85**, **86** on the catalytic activity of ABL1 kinase. While the expected activity of Imatinib and Staurosporine was confirmed; we did not observe any significant inhibitory effect of compounds **85**, **86** on the ABL1 kinase (please, see SI, pages 294, 295; Supplementary Figs. 13–15). The observed results correlate well with the previous study by Nicolaou, Vourloumis, and Stepan who demonstrated that the replacement of the *para*-substituted phenyl ring in Imatinib with various saturated cyclic cores, including bicyclo[1.1.1]pentane and cubane, led to a dramatic loss of potency against the ABL1 kinase[67].

To study the biological activity of Vorinostat and its analogs **88**, **89**, we evaluated their effect on human hepatocellular carcinoma cells HepG2 by fluorescent microscopy (please, see SI, pages 296-300; Supplementary Figs. 16–19). The cells were incubated with the compounds for 48 hours. Staining with specific dyes revealed that all three compounds promoted caspase-dependent cell death, - apoptosis, - that further precipitated in necrosis when the cellular membrane lost its integrity. Vorinostat treatment resulted in 7.2% and 12.2% of apoptotic cells upon incubation at concentrations 5 μM and 50 μM respectively (Fig. 9). Analogs **88** and **89** demonstrated similar efficacy only at 50 μM.

These primary biological results (Fig. 9) suggested that Vorinostat and both its analogs **88**, **89** could have similar cytotoxic and cytostatic

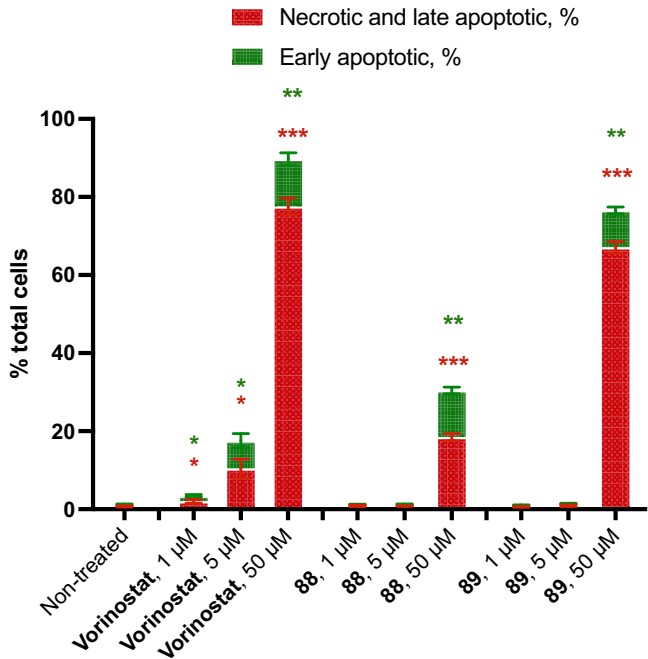

**Fig. 9 | Anticancer activity of Vorinostat (SAHA) and its saturated analogs 88, 89.** Types of HepG2 cell death (% of total cells) after treatment with *Vorinostat* and compounds **88, 89** (1 μM, 5 μM, and 50 μM) for 48 h. Red: necrotic cell death. Green: early apoptotic cell death. The data were presented as mean ± SEM ($n = 3$, independent wells for every of which approx. 2000 visualized cells were analyzed).* - indicates $P < 0.05$, ** - indicates $P < 0.01$, *** - $P < 0.001$ compared with the non-treated group in a two-tailed unpaired t-test with Welch correction on each row of data.

activities in cells (for a more comprehensive comparison of Vorinostat and its analogs **88**, **89**, additional experiments on the enzyme potency and selectivity are needed).

### Virtual libraries

To analyze how the replacement of the *para*-substitued phenyl ring with 2-oxabicyclo[2.2.2]octane affects 3D-shape of organic compounds, we generated two virtual libraries based on *C*- and *N*-terminus modifications of *para*-aminobenzoic acid and its 2-oxabicyclo[2.2.2] octane-containing analog. Each library contained 5000 molecules (Supplementary Data 6, Supplementary Data 7). According to principal moments of inertia (PMI) plots, both libraries occupied essentially the same region in 3D-chemical space. The same was true for FDA-approved drugs Aminopterin, Conivaptan, Deferasifox, Tetracaine, and their 2-oxabicyclo[2.2.2]octane-containing analogs (for details, please see SI, pages 301–304; Supplementary Table 8, Supplementary Figs. 20 and 21).

In conclusion, we have designed, synthesized, and characterized a new saturated bioisostere of the phenyl ring - 2-oxabicyclo[2.2.2] octane. In the design of the structure, we kept all advantages of the previously used cores (bicyclo[1.1.1]pentane, bicyclo[2.2.2]octane, cubane): conformational rigidity, metabolic stability, non-chirality, and collinearity of the exit vectors (Fig. 1c). In addition, we addressed their disadvantages: C-C distance and lipophilicity (Fig. 1c). Thus the 2-oxabicyclo[2.2.2]octane scaffold designed here was synthesized from available starting materials on a multigram scale (Table 1) - up to 135 g in one run (Fig. 2). The key synthesis step was the iodocyclization of cyclohexane-containing alkenyl alcohols with molecular iodine in acetonitrile (Figs. 2 and 3). Crystallographic analysis revealed its high similarity with the *para*-substituted phenyl ring (Fig. 5). 2-Oxabicy-clo[2.2.2]octane core was incorporated into the structure of Imatinib and Vorinostat drugs instead of the *para*-substituted and the *mono*-

substituted phenyl rings, correspondingly (Figs. 7 and 8). In the case of Imatinib, the formed saturated analog **85** possessed improved physi-cochemical properties over the drug: increased water solubility, enhanced metabolic stability, and reduced lipophilicity (Fig. 7). In the case of Vorinostat (SAHA), the formed saturated analog **88** exhibited a similar biological activity compared to that of the drug (Fig. 9).

This study enhances the repertoire of available saturated bioi-sosteres of (hetero)aromatic rings for use in drug discovery projects.

## Methods

### General procedure for the iodocyclization

To a solution of alkene **5** (222.64 g, 1.21 mol, 1.00 equiv) in MeCN (4000 mL) were added $NaHCO_3$ (243.94 g, 2.90 mol, 2.40 equiv) in one portion and $I_2$ (736.60 g, 2.90 mol, 2.40 equiv) in four portions. The resulting mixture was stirred for 12 h at room temperature. Then sodium thiosulfate pentahydrate (900.24 g, 3.63 mol, 3.00 equiv) and distilled water (2000 mL) were added to the mixture. The colorless mixture was extracted with MeO*t*Bu (10 × 400 mL). The combined organic layers were concentrated under reduced pressure to dryness. The residue was dissolved in MeO*t*Bu (1000 mL), washed with brine (1 × 400 mL), a saturated solution of $Na_2S_2O_3$ (3 × 400 mL), dried over $Na_2SO_4$, filtered through a plug of $SiO_2$ (0.5 L glass filter filed with 3 cm in high with silica gel) and concentrated. The final product was purified by column chroma-tography ($SiO_2$, hexane:EtOAc = 1:5, $R_f = 0.7$) to provide pure iodide **6**. Yield: 135.16 g, 0.436 mol, 36%, white solid.

NMR spectra were analyzed with MestreNova (11.0.3-18688).

### Reporting summary

Further information on research design is available in the Nature Portfolio Reporting Summary linked to this article.

## Data availability

Experimental data as well as characterization data for all new com-pounds prepared during these studies are provided in the Supple-mentary Information of this manuscript. The X-ray crystallographic coordinates for compounds **30**, **57**, **67**, **69**, and **78** have been deposited at the Cambridge Crystallographic Data Centre (CCDC) with accession codes 2226162 (**30**), 2226164 (**57**), 2226872 (**67**), 2226163 (**69**), 2266656 (**78**). These data can be obtained free of charge from the Cambridge Crystallographic Data Centre via www.ccdc.cam.ac.uk/structures/. A source data file is available for the biological activity of Imatinib with analogs **85**, **86**; and Vorinostat with analogs **88**, **89**. Source data are provided with this paper.

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

## Acknowledgements

The authors are grateful to Prof. A. A. Tolmachev for the support of this work. This project has received funding from the European Research Council (ERC) under the European Union's Horizon 2020 research and innovation program (grant agreement No. 101000893 - BENOVELTY). P.K.M. is very grateful to Dr. S. Shishkina (IOC, Kyiv) for the X-ray studies, to Dr. D. Bylina for HRMS measurements, to K. Fominova and V. Kokhalskiy for the help with the synthesis of compounds **35** and **39**, and to Dr. V. Kubyshkin for proofreading the manuscript.

## Author contributions

V.V.L., I.S., P.B., and P.K.M. designed the experiments. V.V.L., Y.P., K.S., O.S., V.B., O.S., I.S., L.B., O. K.-U., Y.H., L.L., P.B., K.H., I.B., J.P.B., D.D. conducted and analyzed the experiments described in this report. P.B. and P.K.M. prepared this manuscript for publication.

## Competing interests

The authors declare the following competing interests: V.V.L., Y.P., K.S., O.S., V.B., O.S., I.S., L.B., O.K.-U., Y.H., P.B., K.H., I.B., D.D., P.K.M. are employees of a chemical supplier Enamine. The remaining authors declare no competing interests.
