## [Peer Review File · Nature Communications]

2 Oxabicyclo[2.2.2]octane as a new bioisostere of the phenyl ringReviewers' Comments:

Reviewer #1:

Remarks to the Author:

In this work, the authors reported a beautiful chemistry that enabled modular and scalable synthesis of 2-oxabicyclo[2.2.2]octanes under mild conditions, a bioisostere of the para-substituted phenyl ring. The 2-oxabicyclo[2.2.2]octanes reported in this work structurally feature novel scaffold and substituent diversity. The compounds showed good chemical stability, increased acidity relative to their phenyl counterparts, and improved physicochemical parameters such as solubility, metabolic stability, and lipophilicity. Overall, this chemistry work is well designed and performed. However, the reviewer has additional comments that the authors should consider for further revisions.

Major points:

1. Are the methods reported for the synthesis of 2-oxabicyclo[2.2.2]octanes suitable for imine substrates to give the corresponding 2-azabicyclo[2.2.2]octanes? If the imine substrates work, further structural diversity and a novel bioisostere may be formed. This hypothesis deserves to be examined. If not, related discussion may enrich the chemistry work.
2. The chemical space coverage of the compound library should be analyzed and compared to the FDA-approved drugs. Related discussions are appreciated.
3. The modifications shown in Scheme 6 look fine, but all compounds possess the same 2-oxabicyclo[2.2.2]octane ring, lacking scaffold diversity. It would be better to achieve scaffold diversity based on several representative compounds by coupling other functionalized starting materials or using different reaction conditions. The reviewer think this work is not very challenging (if well designed) considering that the compounds have highly functionalized groups.
4. The data reported in this work do not support the claim that 2-oxabicyclo[2.2.2]octane is a bioisostere of the para-substituted phenyl ring. The authors just compared the physicochemical properties of Imatinib and 2-oxabicyclo[2.2.2]octane containing compound (63), without testing the enzymatic POTENCY AND SELECTIVITY against the targets. The work is not enough to demonstrate that 2-oxabicyclo[2.2.2]octane is a bioisostere of the para-substituted phenyl ring. To support the conclusion, the authors should AT LEAST give two examples, in which the para-substituted phenyl ring is replaced with 2-oxabicyclo[2.2.2]octane. These two examples (approved drugs or clinical candidates preferred) should bind to different targets of interest. Without such biological data, it is inappropriate to conclude that 2-oxabicyclo[2.2.2]octane is a bioisostere of the para-substituted phenyl ring.

Minor points:

1. The title should be modified to "2-Oxabicyclo[2.2.2]octane: A New Bioisoster of the para-substituted Phenyl Ring".
2. The abstract should be rephrased, highlighting the key findings and significance of such ring system for use in drug discovery.
3. In the introduction section, the activities of 2-oxabicyclo[2.2.2]octane containing compounds should be briefly discussed.

Reviewer #2:

Remarks to the Author:

Mykhailiuk and coworkers report on the development of a new class of isosteres for a para substituted aromatic rings. Previous research has shown issues with the more well know classes of isosteres such as bicyclo[1.1.1]pentane and cubane with regards to stability and distances. Bicyclo[2.2.2]octane positions the substituents in the correct environment, yet is too lipophilic for use. This manuscript provides a nice solution to this problem by incorporating an oxygen into the framework. This greatly increases the hydrophilicity of the ring system. In addition, it affects the acidity of a carboxylic to be like an aryl ring. The authors have previously demonstrated a similar effect with a [2.1.1] ring system that was prepared in a similar way (ref 29). Notably, the conditions had to be modified to work for the

[2.2.2] system.

The authors have devised a scalable route to a core scaffold which can be easily diversified. The follow-up analysis and synthesis of a matched pair is well done.

Overall, I am highly supportive of this work. I think the development of new isosteres is very valuable. The authors have made a significant contribution by making structures that are of real utility. Therefore, I support publication pending some minor revisions.

Comments:

There are some odd phrases in the manuscript such as, "After thinking for awhile," and "Being already in a depression". There are more examples like this. This seems a too informal for scientific writing.

Reference 21 is from twitter, is this allowed? Can this type of reference be included?

Can the authors attempt a decarboxylative cross coupling of 43 or 57? It would be nice to aryl group on one side. I think the issue of aryl group incorporation needs to be addressed by some means.

How is phi 1 and 2 not 180 for Imatinib?

Can the work "ideal" be in the title? Ideal means that it is best in all situation. I don't think the authors can claim this.

Reviewer #3:

Remarks to the Author:

This manuscript describes the straightforward synthesis of the oxa-bicyclo[2.2.2]octane motif as a potential bioisostere for para-substituted arenes. In this competitive field, the oxa-BCO architecture is significant as it overcomes limitations in substituent separation (BCP), lipophilicity (BCO) and stability (cubane). As such, this work is sure to be of interest and use to medicinal chemists and agrochemists.

The strategy employed is simple: To effect a scalable synthesis of a exo-methylene cyclohexane bearing a hydroxymethyl (or other alcohol) group at the 4-position that can undergo iodoetherification to cyclise to the oxa-BCO framework. This strategy is already well-established in previous work from Mykhailiuk and co-workers (Ref. 29), and here required only a solvent screen to achieve success. While a drawback in terms of novelty, it is certainly a plus point synthetically.

The chemistry is most directly carried out by a sequence of aldol reaction of an ester at the 4-position with a variety of aldehydes / ketones, then iodocyclization; but notably the identify of the bridgehead substituent (an ester) can be easily 'switched' by reducing the ester such that it becomes the alcohol used for the iodoetherification reaction. This improves the diversity of the substituents that can be installed at the bridgehead. The derivatizations of the iodomethyl group at the other bridgehead into other functional groups are all fairly standard (Scheme 5 is a little repetitive / cluttered in this respect). The main point here is that the BCO can be converted to a variety of 'useful' functional groups through these transforms, rendering the core easy to introduce into other contexts.

These products are impressively stable towards storage, and various conditions (acid, base, heat), demonstrating they should be robust towards a variety of chemical reactions in a synthetic context. Crystallographic analysis demonstrates the overlay of the oxa-BCO with a para-substituted phenyl. Arguably more significant is the characterization of the effect of the oxa-bridge on pKa of acids, and physicochemical and pharmacokinetic properties of imatinib analogues.

In all, I believe this is a highly significant piece of work from the perspective of applications of this core, which has previously only been accessible by quite lengthy or limited routes. The key chemistry itself is however closely related to the authors' previous work. One particular drawback is actually the nature of the alkene – always a methylene group – can other alkenes be tolerated, allowing longer bridgehead chains to be installed? I think it would be useful for the authors to address this question before this work would be suitable for Nature Communications. Another question might be if the iodide is a suitable electrophile for cross-coupling, as it cannot undergo beta-hydride elimination, nor fragmentation as might a small ring structure.

Smaller points:

- Title: 'Bioisoster' should be 'Bioisostere' (and throughout)
 - Abstract: ' and we learn about it already in school' I would suggest to remove this phrase.
 - P1 Col1: ' It is one of the most popular structural motifs in natural products and bioactive compounds' replace 'popular' with 'common' particularly with reference to natural products.
 - p1 Col1: ' However, organic compounds with more than two phenyl rings often suffer from poor solubility and low metabolic stability'. I would expect that a compound with just one phenyl ring could also exhibit low metabolic stability.
- Reference 22: I do not think this is particularly relevant to the current manuscript!
- P2 Col1: ' in the morning'... insert the 12 h instead!
 - P2 Col1: ' The addition of the iodine molecule to the double C=C bond did take place, but the cyclization did not happen.' Does this mean the diiodide was formed? or iodohydrin?
 - P2 Col1: ' Being already in a depression' – remove.
 - P2 Col1: 'we finally tried pure dipolar aprotic solvents' Remove 'pure', could use 'solely' instead.
 - P2 Col1: Table 1 / Scheme 3: What are the side products of this reaction? 36% yield on this scale is impressive – is the rest residual sm, or side products as noted above?
 - P2 Col2: ' Initially, we isolated the transition alcohol', replace 'transition' with 'intermediate'
 - P2 Col2: '3D-shaped' – remove
 - Scheme 4: Replace 'MeCHO' in the header with 'RCHO'
 - Figure 3 – some error in the first structure

Reviewer #1 (Remarks to the Author):

In this work, the authors reported a beautiful chemistry that enabled modular and scalable synthesis of 2-oxabicyclo[2.2.2]octanes under mild conditions, a bioisostere of the para-substituted phenyl ring. The 2-oxabicyclo[2.2.2]octanes reported in this work structurally feature novel scaffold and substituent diversity. The compounds showed good chemical stability, increased acidity relative to their phenyl counterparts, and improved physicochemical parameters such as solubility, metabolic stability, and lipophilicity. Overall, this chemistry work is well designed and performed.

- Thank you!

However, the reviewer has additional comments that the authors should consider for further revisions.

Major points:

1. Are the methods reported for the synthesis of 2-oxabicyclo[2.2.2]octanes suitable for imine substrates to give the corresponding 2-azabicyclo[2.2.2]octanes? If the imine substrates work, further structural diversity and a novel bioisostere may be formed. This hypothesis deserves to be examined. If not, related discussion may enrich the chemistry work.

- Yes, the strategy also works for amine substrates! (amine, right??)

We first synthesized substrate **40** (its preparation was added to the SI). Its cyclization into **41** was not successful neither at rt, nor under a heating (cyclization of analogous OH-substrate also did not work at rt, but did work under heating).

Then, we synthesized the Me-substituted substrate **42** (its full preparation was added to the SI). Cyclization at rt provided product **43** in 31% yield. Under heating, we the yield was improved to 41%.

So, cyclization of *N*-substrates was less effective compared to that of *O*-substrates, but it worked. We have added this material to Scheme 4, the text, and the SI.

2. The chemical space coverage of the compound library should be analyzed and compared to the FDA-approved drugs. Related discussions are appreciated.

- Thank you. We have now performed a full biological evaluation of Imatinib and analogues (analogues are inactive); and Vorinostat and analogues (this is a new material – analogues are active), and added it to the manuscript as a final paragraph. We could add data on the chemical space coverage of O-222, but we just feel that now, given the added biological part, it would be redundant. However, please let us know, that if you still feel that this part is still interesting (in addition to biological experiments) – we will of course do it and add.

3. The modifications shown in Scheme 6 look fine, but all compounds possess the same 2-oxabicyclo[2.2.2]octane ring, lacking scaffold diversity. It would be better to achieve scaffold diversity based on several representative compounds by coupling other functionalized starting materials or using different reaction conditions. The reviewer think this work is not very challenging (if well designed) considering that the compounds have highly functionalized groups.

- Thank you very much for this suggestion. These additional experiments (below) have been performed and added to Scheme 6:

- sulfone **70** (para-Br-aryl scaffold; thiophenol+ iodide **32** and oxydation);
- sulfonamide **71** (thiophene scaffold) - (from amine **48**);
- triazole **72** (quinoline scaffold) - (from azide **62**)

(d) also compounds **75-79** have been made using very different reaction conditions:

- **75** (radical decarboxylative reduction);
- **76** ([Ni]-catalyzed C-C cross coupling);
- **77** (modified Barluenga-Valdes coupling);
- **79** (radical decarboxylative borylation).

4. The data reported in this work do not support the claim that 2-oxabicyclo[2.2.2]octane is a bioisostere of the para-substituted phenyl ring. The authors just compared the physicochemical properties of Imatinib and 2-oxabicyclo[2.2.2]octane containing compound (**63**), without testing the enzymatic POTENCY AND SELECTIVITY against the targets. The work is not enough to demonstrate that 2-oxabicyclo[2.2.2]octane is a bioisostere of the para-substituted phenyl ring. To support the conclusion, the authors should AT LEAST give two examples, in which the para-substituted phenyl ring is replaced with 2-oxabicyclo[2.2.2]octane. These two examples (approved drugs or clinical candidates preferred) should bind to different targets of interest. Without such biological data, it is inappropriate to conclude that 2-oxabicyclo[2.2.2]octane is a bioisostere of the para-substituted phenyl ring.

- We fully agree. Thank you for commenting on that.

We have now additionally tested bioactivity of *Imatinib* and its analogues **85** (O-222), **86** (222) against ABL1 kinase. Both analogues were inactive. This results correlates well with the study of *Nicolauo* and *Stepan* (2016, *ChemMedChem*, 31), where they showed that replacement of the para-phenyl ring in *Imatinib* with bicyclo[1.1.1]pentane and cubane leads to the loss of potency. It seems that *Imatinib* the presence of the para-phenyl ring is crucial for the bioactivity and cannot be replaced with saturated analogues. These results and the discussion have been added to the text and the SI.

<https://chemistry-europe.onlinelibrary.wiley.com/doi/abs/10.1002/cmdc.201500510> (2016, *ChemMedChem*, 31)

In addition, we have also synthesized *Vorinostat* and its analogues **88** and **89**. We also have studied their biological properties (cell death of human hepatocellular carcinoma cells HepG2) – and both analogues were active! We have added this data (Scheme 7) and discussion to the text and the SI.

Minor points:

- The title should be modified to “2-Oxabicyclo[2.2.2]octane: A New Bioisostere of the para-substituted Phenyl Ring”.
 - Thank you, the title has been updated to “2-Oxabicyclo[2.2.2]octane: A New Bioisostere of the Phenyl Ring” .
- The abstract should be rephrased, highlighting the key findings and significance of such ring system for use in drug discovery.
 - Thank you, the abstract has been completely rewritten to put specific findings of this work, and to highlight the importance of this ring system for drug discovery projects.
- In the introduction section, the activities of 2-oxabicyclo[2.2.2]octane containing compounds should be briefly discussed.
 - Thank you. We have added the description of the activities of 2-oxabicyclo[2.2.2]octane containing compounds the introduction section: “Also, 2-oxabicyclo[2.2.2]octane containing molecules exhibited a broad range of biological activities: estrogen receptor-beta agonists,^{18a} myeloperoxidase inhibitors,^{18b} antibacterial agents,^{18c} ¹⁹ DGAT1 Inhibitors,^{20a} and ROR γ t agonists.^{20b}”

Reviewer #2 (Remarks to the Author):

Mykhailiuk and coworkers report on the development of a new class of isosteres for a para substituted aromatic rings. Previous research has shown issues with the more well know classes of isosteres such as bicyclo[1.1.1]pentane and cubane with regards to stability and distances. Bicyclo[2.2.2]octane positions the substituents in the correct environment, yet is too lipophilic for use. This manuscript provides a nice solution to this problem by incorporating an oxygen into the framework. This greatly increases the hydrophilicity of the ring system. In addition, it affects the acidity of a carboxylic to be like an aryl ring. The authors have previously demonstrated a similar effect with a [2.1.1] ring system that was prepared in a similar way (ref 29). Notably, the conditions had to be modified to work for the [2.2.2] system.

The authors have devised a scalable route to a core scaffold which can be easily diversified. The follow-up analysis and synthesis of a matched pair is well done.

Overall, I am highly supportive of this work. I think the development of new isosteres is very valuable. The authors have made a significant contribution by making structures that are of real utility. Therefore, I support publication pending some minor revisions.

- Thank you very much!

Comments:

There are some odd phrases in the manuscript such as, “After thinking for awhile,” and “Being already in a depression”. There are more examples like this. This seems a too informal for scientific writing.

- Right, thank you. The 1st phrase has been replaced with “Considering the possible saturated structures..” The 2^d phrase on a depression has been removed. Also, the language has been revised to become more formal through the text.

Reference 21 is from twitter, is this allowed? Can this type of reference be included?

- To be honest, I do not know. It does not seem to violate any formal rules of Nature journals. But, we have removed it – we agree, it might be too casual for a scientific manuscript.

Can the authors attempt a decarboxylative cross coupling of 43 or 57? It would be nice to aryl group on one side. I think the issue of aryl group incorporation needs to be addressed by some means.

- Right, thank you. The decarboxylative C-C cross-coupling seems to be problematic. We have tried the standard reactions developed by Baran (with Ph₂Zn; *JACS*, 2016, 11132) and Weix (with PhI; *JACS*, 2016, 5016) and did not observe the formation of the requested product.

However, we could perform the decarboxylative reduction (Baran; *ANIE*, 2017, 260) and the decarboxylative borylation (Baran; *ACS Catal.*, 2018, 9537) of these compounds. Synthesis of compounds **74**, **75**, **78** (*X-ray*) and **79** have been added to Scheme 5:

How is phi 1 and 2 not 180 for Imatinib?

- ϕ_1 (Imatinib) = 176° , ϕ_2 (Imatinib) = 178° . This data is included in Figure 2.
It is curious, but these angles also deviate from the ideal 180° .

Can the work "ideal" be in the title? Ideal means that it is best in all situation. I don't think the authors can claim this.

- Right, thank you. The title has been changed to "2-Oxabicyclo[2.2.2]octane: A New Bioisostere of the Phenyl Ring."

Reviewer #3 (Remarks to the Author):

This manuscript describes the straightforward synthesis of the oxa-bicyclo[2.2.2]octane motif as a potential bioisostere for para-substituted arenes. In this competitive field, the oxa-BCO architecture is significant as it overcomes limitations in substituent separation (BCP), lipophilicity (BCO) and stability (cubane). As such, this work is sure to be of interest and use to medicinal chemists and agrochemists.

The strategy employed is simple: To effect a scalable synthesis of a exo-methylene cyclohexane bearing a hydroxymethyl (or other alcohol) group at the 4-position that can undergo iodoetherification to cyclise to the oxa-BCO framework. This strategy is already well-established in previous work from Mykhailiuk and co-workers (Ref. 29), and here required only a solvent screen to achieve success. While a drawback in terms of novelty, it is certainly a plus point synthetically.

The chemistry is most directly carried out by a sequence of aldol reaction of an ester at the 4-position with a variety of aldehydes / ketones, then iodocyclization; but notably the identify of the bridgehead substituent (an ester) can be easily 'switched' by reducing the ester such that it becomes the alcohol used for the iodoetherification reaction. This improves the diversity of the substituents that can be installed at the bridgehead. The derivatizations of the iodomethyl group at the other bridgehead into other functional groups are all fairly standard (Scheme 5 is a little repetitive / cluttered in this respect). The main point here is that the BCO can be converted to a variety of 'useful' functional groups through these transforms, rendering the core easy to introduce into other contexts.

These products are impressively stable towards storage, and various conditions (acid, base, heat), demonstrating they should be robust towards a variety of chemical reactions in a synthetic context. Crystallographic analysis demonstrates the overlay of the oxa-BCO with a para-substituted phenyl. Arguably more significant is the characterization of the effect of the oxa-bridge on pKa of acids, and physicochemical and pharmacokinetic properties of imatinib analogues.

In all, I believe this is a highly significant piece of work from the perspective of applications of this core, which has previously only been accessible by quite lengthy or limited routes. The key chemistry itself is however closely related to the authors' previous work. One particular drawback is actually the nature of the alkene – always a methylene group – can other alkenes be tolerated, allowing longer bridgehead chains to be installed? I think it would be useful for the authors to address this question before this work would be suitable for Nature Communications.

- Thank you for this suggestion. Yes, we have tried two alkenes **34**, **36** substituted at the methylene group. The cyclization worked well, and products **35**, **37**, correspondingly, were obtained. Cyclization of the endocyclic alkene **38**, however, gave an isomeric core, 6-oxabicyclo[3.2.1]octane, **39**. We have added this data to Scheme 4, and the discussion to the text.

Another question might be if the iodide is a suitable electrophile for cross-coupling, as it cannot undergo beta-hydride elimination, nor fragmentation as might a small ring structure.

- Yes, thank you. We have tried the [Ni]-catalyzed C-C cross coupling of the simplest iodide **41** with PhMgBr. Product **76** was isolated in 76% yield. We have added this result to Scheme 4.

Smaller points:

- Title: 'Bioisoster' should be 'Bioisostere' (and throughout)

- We have replaced 'Bioisoster' with 'Bioisostere' everywhere. Thank you.

- Abstract: ' and we learn about it already in school' I would suggest to remove this phrase.

- The phrase has been removed.

- P1 Coll: ' It is one of the most popular structural motifs in natural products and bioactive compounds' replace 'popular' with 'common' particularly with reference to natural products.

- "Popular" has been replaced by "common." Thank you.

- Reference on rings in natural products has been added. Reference 1: Y. Chen, C. Rosenkranz, S. Hirte, J. Kirchmair. Ring systems in natural products: structural diversity, physicochemical properties, and coverage by synthetic compounds. *Nat. Prod. Rep.* **2022**, *39*, 1544-1556.

- p1 Coll: ' However, organic compounds with more than two phenyl rings often suffer from poor solubility and low metabolic stability'. I would expect that a compound with just one phenyl ring could also exhibit low metabolic stability.

- Right, thank you. The phrase has been replaced with the more precise one: "However, organic compounds with more than two phenyl rings often suffer from poor solubility."

Reference 22: I do not think this is particularly relevant to the current manuscript!

- We agree, thank you. We have removed the reference.

- P2 Coll: ' in the morning'... insert the 12 h instead!

- Done.

- P2 Coll: ' The addition of the iodine molecule to the double C=C bond did take place, but the cyclization did not happen.' Does this mean the diiodide was formed? or iodohydrin?

- Formation of the complex mixture was observed with no C=C in the NMR. We think, that iodine added at the C=C bond, and subsequently the compound decomposed (rather than cyclized).

- P2 Coll: ' Being already in a depression' – remove.

- Removed.

- P2 Coll: 'we finally tried pure dipolar aprotic solvents' Remove 'pure', could use 'solely' instead.

- Replaced. Thank you.

- P2 Coll: Table 1 / Scheme 3: What are the side products of this reaction? 36% yield on this scale is impressive – is the rest residual sm, or side products as noted above?

- No, neither SM, nor individual side products - formation of polymeric products.

- P2 Col2: 'Initially, we isolated the transition alcohol', replace 'transition' with 'intermediate'
- Replaced. Thank you.
- P2 Col2: '3D-shaped' – remove
- Removed.
- Scheme 4: Replace 'MeCHO' in the header with 'RCHO'
- This particular example **9** was synthesized from alkene **8** and MeCHO.
- Figure 3 – some error in the first structure
- Sorry for asking – which error exactly? The structure seems to be fine (oxygen atom is red).

Reviewers' Comments:

Reviewer #1:

Remarks to the Author:

The revisions made to the chemistry work have satisfied the reviewer. However, they are not content with the biological part. The use of cell-based assays to validate the conclusion that 2-Oxabicyclo[2.2.2]octane is a new bioisostere of the Phenyl Ring is inappropriate. This is because cellular effects result from multiple factors. To enhance the validity of their findings, the reviewer suggests that the authors include data on enzyme potency and selectivity alongside the reported data.

Considering the structural novelty of the library, it is essential to analyze the chemical space coverage of the compound library. This analysis should be compared to that of the FDA-approved drugs. Such a comparison would be of interest to the broad readership of the journal.

Reviewer #2:

Remarks to the Author:

The authors have addressed my concerns. I now find this paper suitable for publication.

Reviewer #3:

Remarks to the Author:

The authors have done a fine job in addressing all comments of the Referees. I particularly appreciated the inclusion of the azacyclic analogue (and the associated structure limitations), and the use of trisubstituted exocyclic alkenes (and the associated endocyclic limitation). The further inclusion of additional (bioactive) oxa-BCO drug analogues also strengthens the manuscript.

My comment "Figure 3 – some error in the first structure" appears not to apply in the present version – presumably some pdf issue.

In all, I believe this revised manuscript satisfactorily addresses the reviewers' comments and suggestions, and is significantly improved as a result. I recommend acceptance for publication in Nature Communications.

Reviewer #1 (Remarks to the Author):

The revisions made to the chemistry work have satisfied the reviewer. However, they are not content with the biological part. The use of cell-based assays to validate the conclusion that 2-Oxabicyclo[2.2.2]octane is a new bioisostere of the Phenyl Ring is inappropriate. This is because cellular effects result from multiple factors. To enhance the validity of their findings, the reviewer suggests that the authors include data on enzyme potency and selectivity alongside the reported data.

- Thank you, we agree with that.

Unfortunately, our biologists (Petro Borysko, coauthor) say that we need ca. 3 months to deliver the suggested assays from the US (because of the ongoing war, the air borders are closed).

We, however, changed the text in the manuscript to avoid strong statements: *“These primary biological results (Fig. 9) suggested that Vorinostat and both its analogs 88, 89 could have similar cytotoxic and cytostatic activities in cells.”*⁶⁹

Also, we added Ref. 69 with the explanation: *“For more comprehensive comparison of Vorinostat and its analogs 88, 89, additional experiments on the enzyme potency and selectivity are needed.”*

Considering the structural novelty of the library, it is essential to analyze the chemical space coverage of the compound library. This analysis should be compared to that of the FDA-approved drugs. Such a comparison would be of interest to the broad readership of the journal.

- We agree, thank you.

We generated two virtual libraries (5000 molecules in each) based on *N*- and *C*-modifications of *para*-aminobenzoic acid and its 2-oxabicyclo[2.2.2]octane analog. Properties of two libraries were calculated, and PMI-plots were prepared. Also, FDA-approved drugs with the residue of *para*-aminobenzoic acid, - **Aminopterin**, **Conivaptan**, **Deferasifox**, **Tetracaine**, **Mitapivat**, - and their 2-oxabicyclo[2.2.2]octane analogs were added to PMI plots. In brief, two libraries occupied essentially the same region in the chemical space – replacement of the *para*-substituted phenyl ring with 2-oxabicyclo[2.2.2]octane did not affect the 3D-geometry of organic molecules.

On the other hand, the second library (2-oxabicyclo[2.2.2]octane) had better properties: much lower lipophilicity (2.9 vs 4.0), higher Fsp³ index (0.64 vs 0.38), and higher number of natural-like molecules (495 vs 8).

Unfortunately, *Nature Communications* allows only 10 Figures/Tables in the main text, and we had to move some previous material to SI. Therefore, we added a short paragraph on virtual libraries to the text (in green background). And full description on the generation of libraries, their properties, comparison with drugs, PMI-plots was added to SI – pages 301-304 (Supplementary Tab. 8, Supplementary Fig. 20, Supplementary Fig. 21).

Thanks again for the great suggestion. It took us really long to generate those libraries and to run the analysis, but yes – it visually shows an effect of the phenyl replacement with the bioisostere (no change in geometry).

Reviewer #2 (Remarks to the Author):

The authors have addressed my concerns. I now find this paper suitable for publication.

- Thank you!

Reviewer #3 (Remarks to the Author):

The authors have done a fine job in addressing all comments of the Referees. I particularly appreciated the inclusion of the azacyclic analogue (and the associated structure limitations), and the use of trisubstituted exocyclic alkenes (and the associated endocyclic limitation). The further inclusion of additional (bioactive) oxa-BCO drug analogues also strengthens the manuscript.

My comment "Figure 3 – some error in the first structure" appears not to apply in the present version – presumably some pdf issue.

In all, I believe this revised manuscript satisfactorily addresses the reviewers' comments and suggestions, and is significantly improved as a result. I recommend acceptance for publication in Nature Communications.

- Thank you!